# Baculovirus-Free SARS-CoV-2 Virus-like Particle Production in Insect Cells for Rapid Neutralization Assessment

**DOI:** 10.3390/v14102087

**Published:** 2022-09-20

**Authors:** Marcel Jaron, Michael Lehky, Marta Zarà, Chris Nicole Zaydowicz, Aidin Lak, Rico Ballmann, Philip Alexander Heine, Esther Veronika Wenzel, Kai-Thomas Schneider, Federico Bertoglio, Susanne Kempter, Reinhard Wolfgang Köster, Silvia Stella Barbieri, Joop van den Heuvel, Michael Hust, Stefan Dübel, Maren Schubert

**Affiliations:** 1Department of Biotechnology, Technische Universität Braunschweig, Spielmannstraße 7, 38106 Braunschweig, Germany; 2Recombinant Protein Expression Platform, Helmholtz Centre for Infection Research, Inhoffenstraße 7, 38124 Braunschweig, Germany; 3Unit of Brain-Heart Axis, IRCCS Monzino Cardiology Center, Via C. Parea 4, 20138 Milano, Italy; 4Division of Cellular and Molecular Neurobiology, Zoological Institute, Technische Universität Braunschweig, Spielmannstraße 7, 38106 Braunschweig, Germany; 5Institute for Electrical Measurement Science and Fundamental Electrical Engineering, Technische Universität Braunschweig, Hans-Sommer-Straße 66, 38106 Braunschweig, Germany; 6Abcalis GmbH, Inhoffenstraße 7, 38124 Braunschweig, Germany; 7Department of Physics, Ludwig-Maximilians-Universität, Geschwister-Scholl-Platz 1, 80539 München, Germany

**Keywords:** virus-like particles (VLPs), SARS-CoV-2, insect cells, expression vector, antibodies, cellular assay

## Abstract

Virus-like particles (VLPs) resemble authentic virus while not containing any genomic information. Here, we present a fast and powerful method for the production of SARS-CoV-2 VLP in insect cells and the application of these VLPs to evaluate the inhibition capacity of monoclonal antibodies and sera of vaccinated donors. Our method avoids the baculovirus-based approaches commonly used in insect cells by employing direct plasmid transfection to co-express SARS-CoV-2 envelope, membrane, and spike protein that self-assemble into VLPs. After optimization of the expression plasmids and vector ratios, VLPs with an ~145 nm diameter and the typical “Corona” aura were obtained, as confirmed by nanoparticle tracking analysis (NTA) and transmission electron microscopy (TEM). Fusion of the membrane protein to GFP allowed direct quantification of binding inhibition to angiotensin II-converting enzyme 2 (ACE2) on cells by therapeutic antibody candidates or sera from vaccinated individuals. Neither VLP purification nor fluorescent labeling by secondary antibodies are required to perform these flow cytometric assays.

## 1. Introduction

Severe acute respiratory syndrome coronavirus 2 (SARS-CoV-2), first described in December 2019, has caused a still ongoing worldwide pandemic. Biosafety restrictions limit the research with authentic SARS-CoV-2 virus. As a circumvention, scientists developed various pseudovirus systems, e.g., based on vesicular stomatitis virus (VSV) [1,2,3] or lentivirus [4,5,6]. However, the resulting pseudovirus particles typically only express one structural protein of the authentic virus. In the case of SARS-CoV-2, this is typically the spike protein, which provides binding to the human virus receptor angiotensin II-converting enzyme 2 (ACE2). Therefore, the functions of the other structural proteins of SARS-CoV-2 (envelope (E), nucleocapsid (N), and membrane (M) protein) might be overlooked [7]. Further, SARS-CoV-2-derived pseudoviruses still require Biosafety Level 2 (BSL2) safety-level laboratories. An alternative is virus-like particles (VLPs). VLPs can self-assemble upon co-expression of a subset or all viral structural proteins but do not contain genomic information [8]. Hence, they cannot replicate and are considered safe, allowing their production and use in BSL1 laboratories. Further, they can resemble all molecular and morphological features of an authentic virus, and several VLP-based vaccines are approved for clinical use [9].

The production of SARS-CoV-2 VLPs has been described in mammalian cells [10,11,12,13], plant cells [14], yeast [15], and insect cells using the Baculovirus expression vector system (BEVS) [16,17,18]. One of the first SARS-CoV-2 VLP studies was published by Xu et al. in 2020. Xu et al. produced VLP in mammalian cells, finding that E and M protein are critical for VLP formation [11], whereas the influence of N-protein is still debated [12,13]. Kumar et al. studied the VLP formation in Hela cells, allowing monitoring of the SARS-CoV-2 cell entry by a split-Luciferase-based assay [10]. In addition, the expression in plant cells was described [14] and Mazumder et al. produced SARS-CoV-2 VLP in yeast as a promising vaccine candidate [15]. Yet, the most common VLP production system so far is BEVS as it offers generally higher VLP yields than mammalian systems and in parallel can perform complex post-translational modifications [19,20]. Consequently, some VLP produced by BEVS are already approved vaccines for humans, e.g., Cervarix against human papillomavirus [21]. The yield in the case of SARS-CoV-2 VLP by BEVS was reported to be 5.8 × 10^11^ particles per liter [17] while a yield was not reported for other expression systems. VLP produced by BEVS reacted with patient sera and elicited a neutralizing IgG response in the Syrian hamster model of COVID-19, indicating a potential to be used as vaccine regardless of differences to mammalian glycosylation [18].

Despite these successes, baculovirus-based VLP production systems have some major drawbacks. First, adjustment of the ratios between the different proteins is challenging as co-infection by a mixture of baculovirus is not very efficient and generation of co-expressing baculovirus is not simple [22]. Second, baculoviruses are always produced in parallel to VLPs, requiring a tedious and difficult purification process [23]. Third, BEVS is a lytic system, which leads to the release of a large number of potential contaminants that could compromise the quality of the VLPs.

In more recent studies, different plasmid-based expression systems in insect cells have been presented [24,25,26,27,28] that avoid the baculoviral limitations described above. The aim of this study was to establish such a plasmid-based VLP production system in insect cells and in parallel use these VLPs as a cell-based assay for a simple and fast screening of potential protection by sera and monoclonal antibodies. Hereto, we produced SARS-CoV-2 VLP in our baculovirus-free insect cell system [27]. We analyzed different expression vector designs and optimal ratios of the M Protein for our GFP-based cellular assay and compared their performance in spike-dependent ELISA. Afterwards, we assessed the quality of our VLP by transmission electron microscopy (TEM), confocal microscopy, Western blot, and nano tracking analysis (NTA), confirming ACE2 binding and resemblance to authentic SARS-CoV-2 virus. In a next step, we applied these VLPs to establish and optimize our cytometric assay and test the batch-to-batch differences and storage of VLPs. Finally, we screened antibody candidates and sera for their respective inhibition potential using the developed VLP inhibition assay.

## 2. Materials and Methods

### 2.1. Design of Expression Vectors

For transient expression in High Five cells (BNI-TN-5B1-4; Thermo Fisher Scientific, Waltham, MA, USA), the vector pOpiE2 was used [28]. A Kozak sequence was inserted downstream of the OpiE2 promoter and the signal peptide of the mouse Ig heavy chain variable region was added, as this was shown to enhance expression [27]. The E, M, and spike protein (13–1273 aa) encoding DNA sequences of the Wuhan SARS-CoV-2 (GeneBank QIH45025.1, QIH45036.1, and QIH45023.1) were inserted into the expression vector. In addition, a spike protein version stabilized by proline substitutions at position 986 and 987 and “GSAS” substitution at the Furin site (residues 682–685 aa) described by Wrapp et al. [29] was also used. E and spike protein were C-terminal tagged by 6× His and M protein was either fused to GFP11 [30] or full-length eGFP. An overview of the expression vectors can be found in Appendix A.

For transient expression full-length ACE2 without endogenous signal peptide (17–805 aa, NCBI Reference Sequence: NP_001358344.1) expression vector pCSE2.5 in Expi293F cells (A14527; Thermo Fisher Scientific, Waltham, MA, USA) was used. Control cells (not expressing ACE2) were transfected by mock DNA consisting of an unrelated pOpiE2 construct. To obtain cytosolic mCherry expression, pFlpBtM-II-mCherry [31] was added to represent 5% of the total DNA used for transfections. The expression of TMPRSS2 (GenBank: AK313338.1) was achieved by replacing 50% of the pCSE2.5-ACE2 plasmid by pCorona2a1 DNA (coding for TMPRSS2) in the transfection mix.

### 2.2. Cultivation and Transfection of High Five Cells

High Five cells (BNI-TN-5B1-4; Thermo Fisher Scientific, Waltham, MA, USA) were cultivated in EX-CELL 405 medium (Merck, Darmstadt, Germany) at 27 °C and 150 rpm and kept at a cell density of 0.3–5.5 × 10^6^ cells/mL. Transfection was performed as described before [27]. In brief, 4 × 10^6^ cells per mL of transfection volume (typically 30 mL) were centrifuged for 4 min at 180× *g*. The cell pellet was resuspended in fresh EX-CELL 405 media in the respective volume. In total, 1 µg DNA and 4 µg of linear 40 kDa PEI (Polysciences, Warrington, USA) per 1 × 10^6^ cells was added directly to the cell suspension. About 6–10 h after transfection, fresh medium was added to adjust the cell number to ~1 × 10^6^ cells/mL. Then, 48 h after transfection, the volume was doubled and 96 h after transfection, the supernatant containing the VLPs was harvested. Cells were removed by centrifugation at 180× *g* for 4 min. In a next step, supernatant was cleared further by 20 min of centrifugation at 1000× *g*. To concentrate the VLPs, 20% sucrose was added, and the supernatant was centrifuged at 21,000× *g* for 7 h. The resulting pellet was resolved in sterile PBS (pH 7.4) in 1/30 of the original volume.

### 2.3. Purification of VLP

SARS-CoV-2 VLP obtained from 3 L of cultivation supernatant was harvested by centrifugation for 45 min at 3000× *g* and room temperature. After filtration through a 0.2 µm filter (Sartolab-P20 plus, Sartorius, Göttingen, Germany), the supernatant was concentrated approximately 10-fold via diafiltration using a Hollow Fiber Module with a 300 kDa molecular weight cut-off (MidiKros, mPES, 235 cm^2^; Repligen, Waltham, USA) attached to a KrosFlo Research IIi TFF System (Repligen, Waltham, USA). Resulting retentate was filled into ultracentrifuge tubes (No.344058; Beckman Coulter, Krefeld, Germany) and combined with 4 mL of a 20% (*w*/*v*) sucrose cushion in phosphate-buffered saline. Sucrose cushion ultracentrifugation was performed using a Beckman Coulter SW 32 Ti rotor for 2 h at 100,000× *g* and 4 °C. The pelleted material was dried briefly and subsequently resuspended in 200 µL of phosphate-buffered saline. This sample was concentrated to ~75 µL via ultrafiltration by Vivaspin 500 with a 100 kDa molecular weight cut-off (No. VS0142, Sartorius, Göttingen, Germany). For further purification, a linear sucrose density gradient was made through by 3.2 mL of 50%, 40%, 30%, 20%, and 10% (*w*/*v*) sucrose into ultracentrifuge tubes (No. 344061; Beckman Coulter, Krefeld, Germany): To avoid mixing, tubes were frozen in liquid nitrogen before adding the next concentration. The final gradients formed while thawing overnight at 4 °C. The ~75 µL of SARS-CoV2 VLP-containing sample was loaded on top of the gradient. Ultracentrifugation was carried out in a Beckman Coulter SW 32 Ti rotor for 9 h at 100,000× *g* and 4 °C. A Piston Gradient Fractionator (BioComp Instruments, Fredericton, Canada) with an attached BioComp TRIAX Full Spectrum Flow Cell and Gilson FC 203B Fraction Collector (Gilson Inc., Middleton, USA) was utilized to collect 17 fractions of 1 mL each from top to bottom. The sucrose concentrations in the fractions were determined using a Zeiss Abbe-Refractometer (Carl Zeiss AG, Oberkochen, Germany).

### 2.4. SDS-PAGE, In-Gel Fluorescence, and Immunoblotting

Separated gradient fractions were applied to SDS-PAGE for subsequent staining, in-gel fluorescence, and immunoblotting. For SDS-PAGE, pre-cast Mini-PROTEAN TGX Any kD gels (Bio-Rad Laboratories, Hercules, USA) were used. Samples were prepared with reducing 6× Laemmli sample buffer and incubated for 5 min at 37 °C. SDS-PAGE gels were stained using InstantBlue Coomassie Protein Stain (abcam, Cambridge, UK). In gel fluorescence was recorded on a fluorescence scanner FujiFilm FLA-9000 (GE Healthcare, Chicago, USA) set to the EGFP channel. Immunoblotting onto Immobilon-P 0.45 µm PVDF membranes (Merck Millipore, Burlington, VT, USA) was carried out using the Power Blotter Semi-dry Transfer System (Invitrogen, Waltham, MA, USA). Membranes were blocked using 0.05% Tween20 and 5% skim dry milk powder in TBS and incubated for 30 min at 37 °C followed by 30 min at room temperature. After three washing steps with TBST, membranes were incubated overnight at 4 °C with mouse monoclonal Anti-His (C-term)/AP Ab (No. 46-0284, now R932-25) diluted at 1:3000. Following another two washing steps with TBST (15 min) and subsequent equilibration (5 min) with alkaline phosphatase buffer (100 mM Tris-Base, 100 mM NaCl, 5 mM MgCl_2_, pH 9.5), the recombinant 6×His-tagged spike protein was detected by incubation with BCIP/NBT Color Development Substrate (Promega, Madison, WI, USA).

### 2.5. Cultivation and Transfection of Expi293F Cells

Expi293F cells (A14527, Thermo Fisher Scientific, Waltham, MA, USA) were cultivated at 37 °C, 110 rpm, and 5% CO_2_ in Gibco FreeStyle F17 expression media (Thermo Fisher Scientific, Waltham, MA, USA) supplemented with 8 mM glutamine and 0.1% Pluronic F68 (PAN Biotech, Aidenbach, Germany), with the cell density kept at ~0.2–4.5 × 10^6^ cells/mL. Transfection was performed as described before [32]. VLP assays were performed ~48 h after transfection.

### 2.6. Sandwich ELISA

First, 200 ng/well of the Receptor Binding Domain of Spike (RBD) binding human antibody STE90-C11 [33] was immobilized overnight at 4 °C or at RT for ~1 h in a 96-well plate (High Binding, Costar, Corning, NY, USA). Afterwards, wells were blocked by 2% MPBST overnight at 4 °C or for ~1 h at RT, followed by three washing steps using an automatic washing system EL 405 select (BioTek, Winooski, VT, USA) with MilliQ-Tween20 (0.05% *v*/*v*). Next, the concentrated VLP were incubated for 1 h at RT at the indicated dilutions. After another three washing steps as described above, 200 ng/well of soluble ACE2-mFc [32] was added for 1 h at RT. After three washing steps as described above, goat anti-mouse antibody conjugated with HRP (A0168, Sigma-Aldrich, St. Louis, MI, USA, diluted 1:42,000) was added for 1 h at RT. Bound antibodies were visualized by TMB substrate (20 parts TMB solution A (30 mM potassium citrate; 1% (*w*/*v*) citric acid (pH 4.1)) and 1 part TMB solution B (10 mM TMB; 10% (*v*/*v*) acetone; 90% (*v*/*v*) ethanol; 80 mM H_2_O_2_ (30%))). The reaction was stopped by the addition of 1 N H_2_SO_4_ and absorbance at 450 nm with a 620 nm reference was measured in an ELISA plate reader (Epoch, BioTek, Winooski, VT, USA).

### 2.7. VLP Inhibition Cell Assay

In total, 50 µL of 30× concentrated (not purified) VLPs were preincubated for 45–60 min at RT with 50 µL of antibodies or sera or PBS at the indicated concentration or dilution. During that time, transfected Expi293F cells expressing ACE2 or a mock control were counted and 0.5 × 10^6^ cells per well were prepared in 400 µL of FACS buffer (2% FCS, 10 mM EDTA, 1× PBS, pH 7.4) in a 96-deep-well plate (Axygen, Corning, CA, USA). After centrifugation for 4 min at 280× *g*, the supernatant was removed, and the cells were resuspended in the 100 µL of VLP mix and incubated at 37 °C for 60 min before 400 µL of FACS buffer was added. The plate was centrifuged again for 4 min at 280× *g* and supernatant was removed. This was followed by a washing step, resuspending the cells in 400 µL of FACS buffer and centrifugation for 4 min at 280× *g*. The resulting cell pellet was resuspended in 200 µL of FACS buffer and measured in a flow cytometer (MACSQuant, Miltenyi Biotech, Bergisch Gladbach, Germany or BD FACSMelody, BD Biosciences, Franklin Lakes, NJ, USA). Analysis was performed by Flowing Software 2 (Turku Bioscience Centre, Turku, Finland) or FlowJo 10.8.1 (BD Biosciences, Franklin Lakes, NJ, USA) gating single cells and measurement of their GFP median (see Appendix A
for the gating strategy).

### 2.8. Serum Samples

Blood samples were obtained from vaccinated individuals in Germany. All donors were informed about the project and gave their consent for this study. The sampling was performed in accordance with the Declaration of Helsinki. Approval was given by the ethical committee of the Technische Universität Braunschweig (Ethik-Kommission der Fakultät 2 der TU Braunschweig, approval number FV-2020-02). Details of the serum samples can be found in Appendix A.

### 2.9. Transmission Electron Microscopy (TEM)

Transmission electron microscopy (TEM) measurements were carried out using a JEM-1011 (JEOL) microscope operating at 80 kV. To negatively stain the VLP, first, the TEM grid (formvar/carbon-coated copper grid, mesh size 300, Ted Pella, Inc., Redding, CA, USA) was treated in a plasma cleaner for 60 s. Next, 5 µL of the VLP buffer suspension (30× concentrated VLP, not purified) was placed on the TEM gird and incubated for 3 min, followed by a quick washing step and 10 s of staining with 5 µL of 2% *w/v* Uranyl format solution (SPI Supplies, West Chester, PA, USA). Afterwards, the grid was left under the fume hood to thoroughly dry prior to measurements.

### 2.10. Nanoparticle Tracking Analysis (NTA)

The concentration and size distribution of 30× concentrated (not purified) VLPs were measured with NanoSight (NS300) (Malvern Panalytical Ltd., Malvern, UK) equipped with NTA software (version 3.4; Malvern Panalytical Ltd., Malvern, UK). All samples were diluted in PBS to a final volume of 1 mL. Ideal measurement concentrations were found by pre-testing the ideal particle per frame value (20–100 particles/frame). For each sample, 5 videos of 60 s were recorded with a Camer Level of 13. After capture, the videos were analyzed by the NanoSight Software (version 3.4; Malvern Panalytical Ltd., Malvern, UK) with a detection threshold of 5. The settings were established according to the manufacturer’s software manual (NanoSight NS300 User Manual, MAN0541-01-EN-00, 2017).

### 2.11. Epifluorescence Microscopy

Transfected Expi293F cells (all by 5% of pFlpBtM-II-mCherry and 95% of ACE2 or mock expression vector or 47.5% of ACE2 expression vector and 47.5% of pCorona2a1 (encoding TMPRSS2)) were seeded on glass coverslips coated with poly-L-lysine in a 24-well plate. The cells were incubated for 24 h at 37 °C and 5% CO_2_ with 400 µL of F17 media supplemented by 10% FCS. After 24 h, each coverslip was placed upside down on a drop of the indicated VLPs and incubated for 1 h at 37 °C and 5% CO_2_. To fixate the cells, the coverslips were placed in 4% PFA/PBS for 10 min at 4 °C and afterwards washed 5 times with 1× PBS. A drop of mounting media (Fluoro-Gel with Tris Buffer, Electron Microscopy Sciences, Hatfield, PA, USA) was placed on a glass slide and the coverslips were put onto it. Before imaging, the samples were stored for a minimum of one night at 4 °C. The pictures were acquired using an epifluorescence microscope type DM 5500 B from Leica with LAS X software (Leica, Wetzlar, Germany) using a 63× (oil immersion) objective. The used filter for the GFP fluorescence had an excitation range from 460–500 nm and an emission range from 512–542 nm. The used filter for the mCherry fluorescence had an excitation range from 542–582 nm and an emission range from 602–644 nm. Exposure times ranged from 134 to 705 ms and the gain was set to 1.0. The scale bars were edited using FIJi software.

### 2.12. Confocal Laser Scanning Microscopy

The samples were prepared as described for the epifluorescence microscopy. The pictures were acquired using a confocal laser scanning microscope type TCS SP8 DMI 6000 from Leica with LAS X software (Leica, Wetzlar, Germany) using a 63× (water immersion) objective. The used laser for the GFP fluorescence had an excitation of 476 nm, an intensity of 10%, and the emission ranged from 488–556 nm. The used laser for the mCherry fluorescence had an excitation of 561 nm, an intensity of 4.9%, and emission ranged from 566–779 nm. The pictures had a format of 2048 × 2048 and were taken with a speed off 600–700 Hz and a line average of 2–3.

## 3. Results

### 3.1. Expression Vector Design and Ratio

For the expression of SARS-CoV-2 VLP E, M, and spike protein were co-expressed. In SARS-CoV-2, the spike protein is preceded by a signal peptide. We exchanged the viral signal peptide for the mouse Ig heavy chain variable region signal peptide (“SP”) shown to work well in High Five cells [27]. Furthermore, we compared VLP containing the wildtype full-length spike protein to VLP containing full-length stabilized spike version (with proline substitutions at position 986 and 987 and “GSAS” substitution at the Furin site, residues 682–685 aa as described in Wrapp et al. [29]). E and M protein of SARS-CoV-2 are not predicted to contain a signal peptide but are still targeted to the endoplasmic reticulum in mammalian cells [34]. Thus, E and M protein expression vectors with and without signal peptide were tested to elucidate whether membrane expression in insect cells could be enhanced. Nucleocapsid (N) protein was not included as it is not essential for VLP formation [10].

After baculovirus-free production, the enriched VLP were analyzed by sandwich ELISA using the SARS-CoV-2 spike RBD-specific antibody STE90-C11 [33] for capture and soluble ACE2-mFc fusions for detection (Figure 1A). For initial analysis, a 1:1:1 ratio of S:E:M expression vectors was used. When the signal peptide was fused to M protein, the ELISA signal of VLP binding decreased significantly. In comparison, the fusion of the signal peptide to the E protein increased binding. As a result, all VLP expression vectors used in the subsequent studies encoded the spike and E protein preceded by the signal peptide while the M protein was expressed without one.

A main advantage of the plasmid-based system is the opportunity for a straightforward adjustment of plasmid ratios, as up to 100,000 plasmids can be taken up by a single cell at least for the calcium-based transfection method [35], suggesting similar levels for the PEI-based transfections. Here, we aimed to optimize the M protein-eGFP (full-length) fusion expression, as this determines the maximal assay signal, while at the same time, an appropriate amount of spike protein on the VLPs has to be maintained. Thus, different ratios (1:1:1 (VLP-1M), 1:1:4 (VLP-4M), 1:1:6 (VLP-6M), and 1:1:8 (VLP-8M)) of S:E:M expression vector and a VLP-6M version that does contain wildtype spike (VLP-6M-Furin) were analyzed by sandwich ELISA. The SARS-CoV-2 spike RBD-specific antibody STE90-C11 was used for capture and soluble ACE2-mFc fusions for the detection of the VLPs (Figure 1B). No differences between the VLP S:E:M ratios were observed with the exception of VLP-6M-Furin, which only showed slight binding at the highest concentration used. In addition, flow cytometry was used to determine the amount of VLP bound to ACE2-expressing Expi293F cells (Figure 1C). Again, no significant differences were observed. Surprisingly, even VLP-6M-Furin led only to a tendentially lower signal. For the subsequent cell binding assays, VLP-6M was chosen.

### 3.2. Quality of SARS-CoV-2 VLP

The VLP size, morphology, quality, and size distribution were assessed by negatively stained transmission electron microscopy (TEM) and nanoparticle tracking analysis (NTA) (Figure 2). The VLPs showed the expected structure in TEM and were similar to authentic SARS-CoV-2 virus [36]. The amount of spike protein visible on the VLP surface correlates to the amount of spike expression vector applied in transfection, resulting in a typical spike aura for VLP-1M, VLP-4M, and VLP-6M but a less obvious one of the other versions. The yield per liter of culture supernatant was determined by NTA to be around 10^13^ particles for all variants except for VLP-6M-Furin (7.6 × 10^12^). Furthermore, the NTA revealed average diameters ranging from 123 (VLP-8M) to 145 nm (VLP-6M), which is in line with the diameter of coronaviruses of 100–200 nm [37]. In addition, VLP-6M was purified by sucrose gradient centrifugation and the incorporation of spike and M protein was confirmed by SDS PAGE and Western blot analysis whereas E protein was not detectable (see Appendix A).

### 3.3. Visualization of VLP Binding to Cells

SARS-CoV-2 virus binds to the ACE2 receptor and can enter the cells via the Furin and TMPRSS2-dependent so-called “early pathway” or the less efficient “late pathway”, which is described to be Furin and TMPRSS2 independent [38]. Therefore, VLPs with and without a Furin site in the spike protein can, in theory, be taken up by the cells, with a higher uptake rate expected for cells expressing both ACE2 and TMPRSS2 and VLPs still containing the Furin site. By confocal microscopy, we analyzed the location of concentrated VLPs containing the stabilized spike protein (VLP-6M) or containing the wildtype spike (VLP-6M-Furin) on cells either not expressing ACE2, expressing ACE2, or expressing ACE2 and TMPRSS2 (Figure 3A, Appendix A).

VLP were visible as small green dots on the cells due to their M protein-eGFP content in all images, but a high laser intensity had to be applied that resulted in auto-fluorescence of the cells. When cells did not express ACE2, VLPs could not be detected. On Expi293F HEK cells expressing ACE2 or coexpressing ACE2 and TMPRSS2, VLPs were located mainly on the surface of the cells. Co-localization with mCherry expressed in the cytoplasm could be observed for some individual ACE2- and TMPRSS2-positive cells with VLP-6M-Furin (see Appendix A). The GFP intensity and number of observed VLPs seemed to be slightly higher when VLP-6M was used compared to VLP-6M-Furin while co-expression of TMPRSS2 did not seem to have an influence. Yet, flow cytometric analysis revealed a significant difference between the VLP and GFP histogram of VLP-6M, which resulted in two distinct populations (Figure 3B, green). One population expressed ACE2 and bound VLP-6M while the other did not. In contrast, for VLP-6M-Furin, a shift in the population was observed (Figure 3B, violet), reflecting the lower binding capacity shown in ELISA (Figure 1B). Coexpression of TMPRSS2 did not lead to a significant difference in the signal to noise ratios obtained. Only a slight tendency for a reduction in the signal to noise could be observed for VLP-6M-Furin binding to ACE2 positive cells vs. ACE2 and TMPRSS2 positive cells (see Appendix A), suggesting, similar to the microscope results, a more efficient uptake of these VLPs.

### 3.4. Optimizing of the Cell Binding Assay

To optimize the signal to noise ratio of VLP binding to the ACE2 positive cells compared to ACE2 negative cells in flow cytometry, different incubation temperatures (4 or 37 °C) and times (60, 120, 180 min; Figure 4A) were tested. Incubation of VLPs and cells at 37 °C led to a significantly higher signal to noise ratio and lower standard deviation than incubation at 4 °C, independent of the incubation time. After 2 h, the highest signal to noise (~3.2) was obtained while after 1 or 3 h, a signal to noise ratio of ~3 could be observed, suggesting that the assay could be performed within an hour without significant loss of signal.

The stability of the VLPs was monitored after storage of the concentrated VLPs at different temperatures over 4 weeks (−80 °C, 4 °C, 37 °C, Figure 4B). The signal to noise ratio was always higher for VLPs stored at −80 °C compared to 37 °C and at least slightly higher compared to storage at 4 °C, suggesting −80 °C as the optimal storage temperature for VLPs. Interestingly, individual Expi293F HEK transfection batches seemed to influence the result more than storage itself, despite a relatively constant transfection efficacy of around 65–75% for all experiments.

Finally, different VLP batches of the same composition (VLP-6M) were compared. A batch-to-batch variation was observed, with two batches (batch#1 and batch#5) providing a signal to noise ratio of ~1.5 whereas all other tested batches provided ~2.5 (Figure 4C). In addition, the absolute GFP value differed between the batches and a higher GFP signal correlated with a higher background signal (Figure 4D). For example, batch#4 provided the highest absolute GFP signal but the signal to noise of batch#2 was in a similar range while it just reached half of the absolute GFP signal. Yet, and most importantly, the GFP signal of all batches varied significantly on ACE2 positive cells compared to cells transfected by mock DNA. Interestingly, no correlation between the total protein amount to the obtained signal to noise or to the total GFP signal was observed but the spike amount seemed very low in batch#1 und batch#5 (see Appendix A).

In conclusion, VLP-6M was suitable to provide a significant GFP signal on ACE2 positive Expi293F cells without purification (only concentration) or staining steps being required. Yet, for direct comparisons of different antibodies or sera with respect to their neutralization capacity, it is recommended to use the same batch of VLPs.

### 3.5. Neutralization Ranking of Monoclonal Antibodies and Sera

The influence of neutralizing antibodies on VLP cell binding was tested with both recombinant monoclonal antibodies and sera of vaccinated donors (Figure 5). Titration of soluble ACE2-hFc or STE90-C11 (an antibody against RBD in clinical trials [33]) showed a successful concentration-dependent inhibition of the VLP binding to the cells, with STE90-C11 being more efficient, probably due to its higher affinity (560 pM) compared to ~20 nM of ACE2 (Figure 5A). To assess the potential of the assay to provide a quick ranking of the neutralization potential, we tested various anti-spike protein monoclonal antibodies at a fixed concentration of 150 µg/mL (Figure 5B). The neutralization potential of these antibodies against authentic virus and their performance in a soluble spike protein inhibition assay is known [33]. The ranking by VLP cell binding identified the identical four best candidates (STE90-C11, STE90-B2-D12, STE94-F12, and STE94-B1-E12) independent of the method (Table 1). The VLP assay mirrored the results of the authentic virus neutralization assay more closely than the soluble spike inhibition assay. Most significantly, the VLP cell binding assay is simpler and more rapid as it does not require a purification step and no antibody staining.

Furthermore, we analyzed the inhibition of VLP binding to cells by sera of vaccinated persons in comparison to pre-SARS-CoV-2 sera. Here, a significant inhibition was only achieved at the lowest dilution (1:10) with sera of a 3× BNT162b2 person (Figure 5C). Sera of four individuals (single vaccinated with Ad26.COV2.S, double with BNT162b2, or three times with BNT162b2) were tested for inhibition at dilutions of 1:20 (Figure 5D) or 1:50 (Figure 5E). For both dilutions, the best inhibition was achieved with the serum from 3× BNT162b2 vaccinated donors. This confirms the importance of a third “booster” vaccination is in accordance with previous results [39].

## 4. Discussion

In this study, we successfully applied baculovirus-free protein expression in insect cells for the production of SARS-CoV-2 VLPs. The baculovirus-free approach allowed for an easy optimization of expression vector combinations and ratios, thereby promising rapid adaption of the VLPs to the latest SARS-CoV-2 variant as it only requires the exchange of one of the vectors. The yield of unpurified VLPs was close to 10^13^ per L culture, promising a higher yield of purified VLPs than reported by Naskalska et al. [17], who obtained 5.8 × 10^11^ particles per L after purification using a BEVS system and claiming that as a high yield. The presumably higher yield obtained in the plasmid-based system compared to BEVS, despite the availability of stronger very late promotors (p10 or polH) in BEVS, might be explained by the fact that BEVS simultaneously produces baculoviral particles and is a lytic system. Thus, translation machinery is not only occupied to produce VLPs but also baculovirus particles in parallel. Additionally, the time frame to have optimal expression and functional secretion of VLPs before cell death is narrow. In addition, expression vectors and their ratios have been adjusted here, which was not the case and is not feasible in BEVS. Comparison to the mammalian production system is not conclusive as to our knowledge, none of the published studies report the obtained amount of SARS-CoV-2 VLP per L so far. Yet, soluble fragments of spike S1–S2 and S1 alone were produced in a much lower amount (<8× or <15× less, respectively) in Expi293F cells compared to the yield obtained in the plasmid-based High Five cell system [32], suggesting a lower rate of VLP production in the mammalian system. This might be due to the simpler and more homogenous glycosylation in insect cells but is still an unexpected observation for a virus infecting mammalian cells. To reliably determine the best system for SARS-CoV-2 VLP production, a comparison of all systems side by side before (or after) purification using the same analysis methods (e.g., NTA) to avoid bias would be required. Furthermore, in future studies, the VLP yield in the baculovirus-free system may be increased by co-expression of N protein. Despite the fact that N protein may not be incorporated into the VLP [17], it was found to considerably enhance the production of VLP in HEK cells [10].

A high similarity between the VLP and authentic SARS-CoV-2 virus was observed in terms of the microscopic appearance, binding to ACE2, and diameter. While the presence of E protein could not be directly confirmed, which might be caused by its small size and general low amount of incorporation into SARS-CoV-2 virions, its presence may be assumed from its main function to help with the assembly of the virions [40]. Additionally, comparison of the E protein expression vectors with and without signal peptide had an influence on the total ELISA signal, again hinting at successful expression of the E protein (Figure 1A). Differences between the insect cell-derived VLPs and mammalian cell-derived VLPs or authentic virus can be assumed to be the different glycosylation and the insect-specific lipid composition of the membrane.

Furthermore, as our main aim was to apply the VLPs for rapid, scalable, parallelized inhibition assays compatible with BSL1 conditions, we used a stabilized variant of the spike protein to enhance binding to ACE2. This stabilized variant led to much higher signals in spike-dependent sandwich ELISA than the wildtype version, which only showed a background signal in ELISA (Figure 1B, comparison of VLP-6M and VLP-6M-Furin). Additionally, the cell populations of binding VLP and not binding VLP cells could not be clearly distinguished in the cytometric assay for VLP-6M-Furin compared to VLP-6M (Figure 3B). In contrast to that, the signal to noise ratio in the cytometric assay was not significantly affected by this and only tendentially decreased (Figure 1C), an observation that correlates with the slightly but significantly lower amount of VLP-6M-Furin compared to VLP-6M (Figure 2C). A possible explanation for the differences in the ELISA performance to the obtained signal to noise ratio is the processed spike. As High Five cells process the Furin site of S1 by around 50% [32], the same can be assumed for full-length wildtype spike. In addition, the diameter of VLP-6M-Furin is reduced (Figure 2D), also indicating a lower number of unprocessed full-length spike on the surface of these VLPs, which was also hinted by the TEM results (Figure 2A). Yet, only the full-length spike version contains the RBD and thereby the ability to bind ACE2. In ELISA, the amount of full-length spike might not be enough to make the VLP-6M-Furin stick to the ACE2 immobilized on a plate during the stringent washing procedure that was performed three times, whereas the VLP-6M has a higher avidity (as it contains only full-length spike) and thereby survives the stringent washing steps. In comparison, the washing procedure in the cytometric assay is much less stringent (careful washing by hand, only one step) and ACE2 might be presented more naturally than when it is immobilized on a plate. Thus, VLP-6M-Furin (but probably in a lower amount; compare Figure 3B) remain bound to the ACE positive cells with less background on ACE2 negative cells, which may explain the similar performance in respect to the signal to noise ratios. In conclusion, the VLP-6M-Furin (able to employ both entry pathways) could be used to set up another assay for analysis of the VLP entry into cells by employing another reporter system activated only upon cell entry. Such a system could be applied to identify antibodies that do not directly bind the RBD of spike but block the entry into the cells by other mechanisms.

In general, our VLPs can be used in simple and rapid cell binding assays amenable to the 96-well plate format to screen for virus-inhibiting antibody candidates, reflecting authentic virus neutralization. No VLP purification—just concentration by simple centrifugation—and no labeling or antibody staining is required for the assay. Future improvements of the VLP should include an exchange of the eGFP to, i.e., a red fluorescence marker that does not overlap with cellular autofluorescence [41].

In conclusion, our study highlights the potential of baculovirus-free expression in insect cells not only to rapidly generate multi-protein VLPs of SARS-CoV-2 but also for other viruses. The approach further offers a new opportunity for vaccine production.

## Figures and Tables

**Figure 1 viruses-14-02087-f001:**
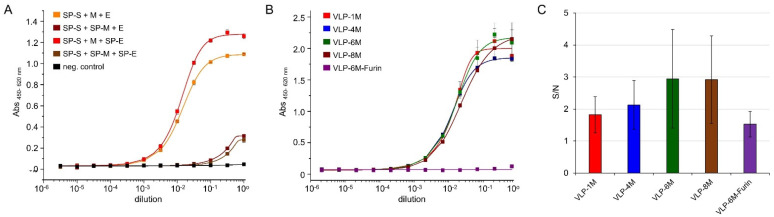
Spike protein and GFP expression analysis for different vector design and ratios. (**A**) Sandwich ELISA of concentrated VLP expressed by co-transfection of the indicated vectors. For VLP capturing, anti-RBD antibody STE90-C11 was coated and soluble ACE2-mFc fusion was used for detection. M protein was fused to GFP11. Technical triplicates were measured, and standard deviation is indicated. Curves were fitted using the Logistic5 function in OriginPro 2019b. SP stands for signal peptide of the mouse Ig heavy chain variable region. (**B**) Influence of different ratios of spike, E, and M-eGFP expression vector on VLP production measured in sandwich ELISA of concentrated VLPs. VLPs were expressed by co-transfection of the vectors in the indicated ratios using either stabilized spike protein version or wildtype (VLP-6M-Furin). For VLP capture, anti-spike antibody STE90-C11 was coated and soluble ACE2-mFc fusion was used for detection. Error bars indicate the standard deviation of technical triplicates. Curves were fitted using the Logistic5 function in OriginPro 2019b. (**C**) GFP median signal to noise ratios of the indicated GFP-positive VLPs binding to recombinantly ACE2-expressing Expi293F cells versus ACE2-negative cells determined by cytofluorometry.

**Figure 2 viruses-14-02087-f002:**
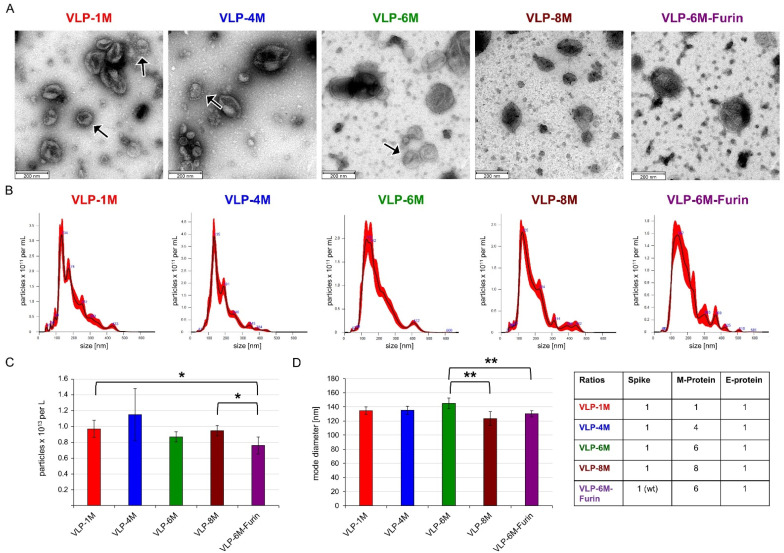
TEM pictures and NTA size analysis of the concentrated VLPs (not purified) with different ratios of spike, E, and M proteins. (**A**) TEM pictures of the indicated VLPs. Scale bar is 200 nm. Arrows indicate typical Spike Corona aura (**B**) Representative NanoSight traces of the indicated VLPs. (**C**) VLP yield per L of culture assessed by NTA. Error bars indicate the standard deviation between three different production batches. (**D**) Mode diameter of the VLPs. Error bars indicate the standard deviation between at least three different production batches of VLPs. Significances were determined by two-sided *t*-Test (* = 90%, ** = 95%). If not indicated, no significant differences were observed.

**Figure 3 viruses-14-02087-f003:**
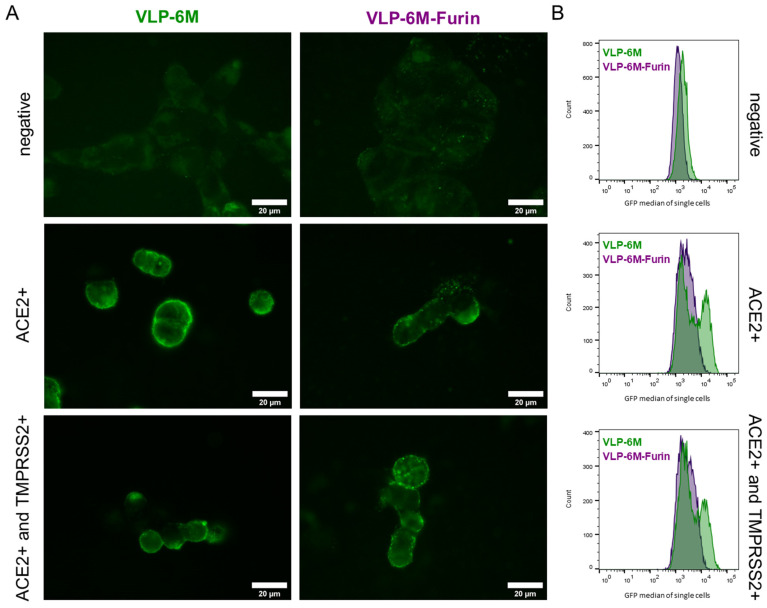
Visualized binding and localization of VLP with stabilized (VLP-6M) or wildtype spike protein (VLP-6M-Furin). (**A**) Confocal GFP images of VLP-6M and VLP-6M-Furin on ACE2 negative, ACE2 expressing, and ACE2 and TMPRSS2 co-expressing Expi293F HEK cells. VLPs are visible as small green dots. Bright field, red fluorescence (=mCherry), and merged images can be found in Appendix A. (**B**) GFP histograms of the indicated VLP binding to ACE2 negative cells, ACE2 or ACE2, and TMPRSS2 expressing Expi293F cells.

**Figure 4 viruses-14-02087-f004:**
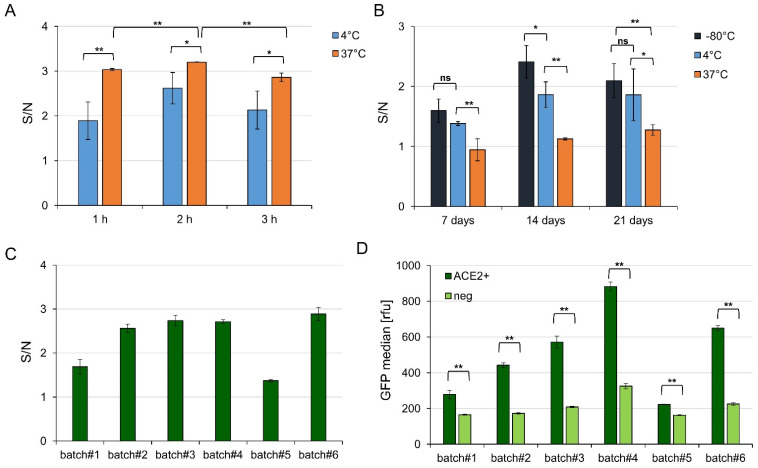
Influence of incubation time, VLP storage, and batch-to batch variation on VLP binding to ACE2 transfected cells. (**A**) Signal to noise ratios of VLP binding to recombinant ACE2 expressing Expi293F cells versus ACE2 negative cells determined by cytofluorometry of one batch of VLP-6M (GFP positive) after 1, 2, or 3 h of incubation at 4 or 37 °C. Error bars indicate the standard deviation between three different independently transfected Expi293F cell preparations. (**B**) Signal to noise ratios of VLP binding to recombinant ACE2 expressing Expi293F cells versus ACE2 negative cells determined by cytofluorometry of one batch of VLP containing GFP-fusions after storage at −80, 4, or 37 °C for 7, 14, or 21 days. Error bars indicate the standard deviation between three different independently transfected Expi293F cell populations. (**C**) Signal to noise ratios of different VLP production batches (standard deviation represents technical triplicates) and (**D**) their respective absolute GFP signals on ACE positive or mock DNA transfected Expi293F cells. Significances were determined by two-sided *t*-Test (* = 90%, ** = 95%).

**Figure 5 viruses-14-02087-f005:**
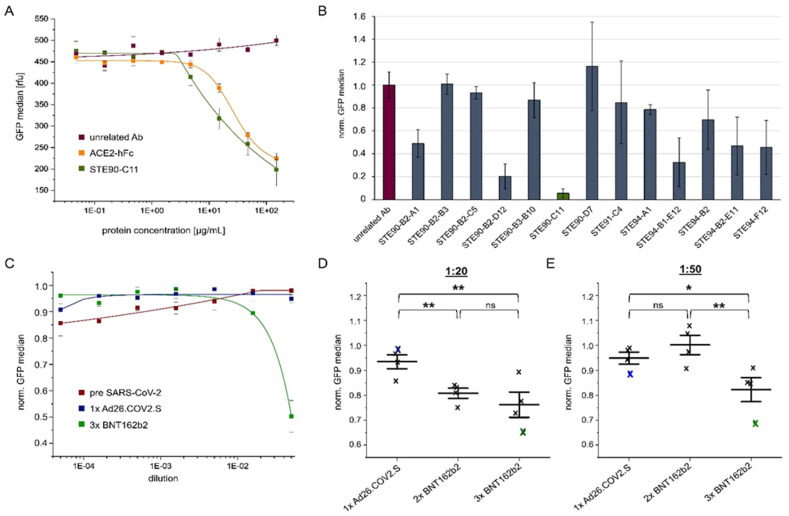
Analysis of anti-spike antibody candidates, ACE2-hFc, or sera by cellular VLP binding assay to measure their respective inhibition potential. (**A**) Absolute GFP values when unrelated antibody (Ab), ACE2-hFc, or STE90-C11 (IgG) were applied for inhibition of the VLP binding to ACE2 positive cells at the indicated concentrations. (**B**) Obtained inhibition of VLP binding using different antibody candidates at a concentration of 150 µg/mL. Bars represent the normalized binding activity by setting the VLP binding without antibody to 1 and binding to ACE2 negative cells to 0. (**C**) Inhibition of VLP cell binding by human sera from, respectively, one pre-SARS-CoV-2 donor or individuals vaccinated with 1× Ad26.COV2.S or 3× BNT162b2. Values represent the binding activity normalized by setting the signal of VLP binding to ACE2 positive cells without sera to 1 and binding to ACE2 negative cells to 0. (**D**) VLP cell binding inhibition by sera from 1× Ad26.COV2.S, 2× BNT162b2, or 3× BNT162b2 vaccinated donors at a dilution of 1:20. The blue or green crosses identify the sera used in (**C**). (**E**) Inhibition of the same sera used in B at a dilution of 1:50. The blue or green crosses identify the sera used in (**C**). All experiments were performed in triplicates using one batch of VLP but three independent batches of ACE2 (or mock) transfected Expi293F cells. The standard deviation is indicated by error bars. Significances were determined by two-sided *t*-Test (* = 90%, ** = 95%).

**Table 1 viruses-14-02087-t001:** Heatmap comparison of VLP cell binding inhibition assay to the inhibition of soluble spike protein binding to ACE2 positive cells and authentic virus neutralization. Data of authentic virus neutralization and binding of trimeric spike protein to ACE2 positive cells were taken from Bertoglio et al. [31] for comparison. Color code: From dark green to yellow to dark red = from the highest neutralization potential to the lowest potential in the respective assay.

Antibody Clone	Neutralization of Authentic SARS-CoV-2 Virus (Plaque Assay) [31]	Trimeric Spike Protein Cell Binding Inhibition [31]	VLP Cell Binding Inhibition Assay
STE90-D7	0.00	0.33	−0.16
STE94-B2	0.06	0.57	0.30
STE90-B2-C5	0.13	0.59	0,07
STE91-C4	0.14	0.75	0.15
STE90-B3-B10	0.18	0.65	0.13
STE90-B2-B3	0.22	0.36	−0.01
STE90-B2-A1	0.25	0.59	0.51
STE94-A1	0.32	0.58	0.21
STE94-B1-E12	0.50	0.88	0.68
STE94-F12	0.65	0.82	0.54
STE90-B2-D12	0.89	0.92	0.80
STE90-C11	0.98	0.98	0.95

## Data Availability

All primary data analyzed in this study will be made available upon reasonable request to the corresponding author.

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
