# Peer review of "Baculovirus-Free SARS-CoV-2 Virus-like Particle Production in Insect Cells for Rapid Neutralization Assessment"

_viruses, 2022, doi:10.3390/v14102087_

Round 1

Reviewer 1 Report

This paper makes a useful contribution by showing that secreted VLP’s (Virus like particles) can be produced using insect technology without requiring the use of recombinant baculoviruses. They show that VLP’s made up of more than one virus protein can be produced by employing direct plasmid transfection which has the following advantages:

1   1. The ratio of individual proteins of the VLP that are expressed can be readily manipulated by controlling the amount of plasmid transfected into the cells for each virus gene of interest. Controlling such ratios using separate recombinant baculoviruses to express each gene or constructing a single recombinant baculovirus to express multiple genes requires more effort.

2   2. Purification of the VLP’s is simpler as no baculovirus budded virus is produced which can complicate the purification of secreted VLP’s.

The paper shows that the VLP’s produced are of a decent quality in terms of reflecting similar structures and size to the native SARS-COV-2 virus and the VLP’s can be used as a safe substitute to other approaches in developing SARS-COV-2 neutralization assays and possibly for producing a SARS-COV-2 vaccine.

It would be useful if the authors can comment more on why such an approach may work better in insect cells than in other expression systems such as mammalian, yeast and plant expression systems. Why does this approach (plasmid transfection) to make VLP’s for SARS-COV-2 (a mammalian virus), work better in insect cells rather than in mammalian cells for example. Perhaps it is because the glycosylation system is simpler in insect cells and so yields of VLP are better and the exact glycosylation is not important for VLP formation and assay/vaccine performance? I accept that the authors may not know the reason but speculation as to the reason can be useful I think.

Also why are yields of secreted VLP’s as good as or better than when recombinant baculoviruses are used with insect cells to produce similar VLP’s, given that stronger promoters are used when recombinant baculoviruses are employed to express the individual virus genes. Perhaps this is because the VLP secreted yield is limited by the protein processing processes in the ER etc and not by the level of transcription/translation of the individual virus proteins?

A list below is made of specific issues that the authors should address and a copy of their paper with edits is attached. Most edits in the attached document are covered also in the list below, although some minor English corrections and cases where the wrong sup figures are referred to are also indicated in the attached edited copy of the paper.

      1. Define ACE2, line 28. This is defined in the introduction but needs to be defined in the abstract also.

      2. Line 113, Useful to indicate a typical transfection volume used here

3    3. Line 171, Define the RBD term - receptor binding domain of the spike protein?

4    4. Line 185, Concentrated VLP? VLP concentration in particles/ml? Is it necessary to standardize the number of VLP particles/ml for the assay procedure?

5  5. Line 210, Purified VLP or concentrated VLP? Was the VLP particles/ml concentration known for samples analyzed?

6    6. Line 286, Ref 35 refers to Ca Phosphate transfection of mammalian cells while in this paper you are transfecting insect cells using PEI, so is this the best ref to support the fact you are getting 100,000 plasmids taken up by a single cell? I am sure you are correct - each cell is getting a lot of plasmid copies but explain why you believe this is an appropriate ref here.

7    7. Line 286, Which GFP fusion was used - GFP11 or full length GFP?

8    8. Line 296, Not clear what you mean by Tendentially here? VLP-6M-Furin data in Fig 1 B and 1 C do not seem to be consistent and deserve further discussion.

9   9. Line 310, Supp Fig 2, not supp Fig 1. Do you refer to Supp Fig 1 in the text? Original images do not have the same Fig number as the corresponding figures in the paper.

1   10. Line 312, Useful to use an arrow in Fig 2 A to show the possible ring of Spike protein around some of the VLPs. The spike proteins are not as obvious as in figure 2 shown in your ref number 36. At least for your VLP-1M and VLP-4M photos you have some evidence in your EM photos of a ring of Spike proteins around your VLP's but these are not so clear in the VLP-6M and VLP-8M photos. You need discuss these images in more detail to convince the reader that you have VLP's displaying spikes as expected for these VLP's.

1   11. Line 344, Why are there 2 distinct populations/peaks in Fig 3B, VLP-6M, ACE2+ cells? What are the units of the X axis in this figure? Explain what you mean by the Fig 3B results reflect the Fig 1B ELISA results.

1    12. Line 416, blue not orange cross?

1    13. Line 439-441, Discuss in more detail why you believe your VLP yield is higher than that in ref 17. Did ref 17 report a yield/L of culture and report their purification yield?

1   14. Line 446-449, Discuss in more detail the evidence available to suggest the E protein is essential for VLP formation.

Reviewer 2 Report

The manuscript describes the production of SARS-CoV-2 virus like particles (VLP) in insect cells by means of plasmid transfection, avoiding the formation of baculovirus particles as contaminating by-product. Individual expression plasmids encoding membrane, envelope and spike protein were transfected into High-Five cells at optimized ratios and self-assembled into authentic structures that resemble their native counterparts. Size and shape of produced VLP were confirmed by electron microscopy and nanoparticle tracking analysis. VLP than were used in cell based assays to test for neutralizing properties of sera from vaccinated individuals and inhibition capacities of monoclonal antibodies.

The report at hand is well written and easy to follow. In the introduction the authors give an overview that covers alternative systems to produce Sars-CoV-2 VLP as well as pseudovirus systems and comment on the relevance of biosafety issues. The insect cell/baculovirus system was chosen due to its established status and high yield expectations as compared to mammalian cells. High yields might be achieved in the classical baculovirus infection process, but for plasmid based transfection, this is questionable (I will come to that later).

Results are clearly structured and are arranged consistently. There is a logical flow starting with production, quality control, binding assay, optimization and inhibition assay for antibodies and sera. I would like to emphasize that optimization of VLP binding to ACE-receptor was done carefully if out of necessity, since signal to noise ratios are very low. On the other hand, batch to batch variation is a critical point, and many researchers are not fully aware of.

In the material section the vector design is described. A graphical illustration would be helpful. VLP purification as outlined here is extremely tedious including several filtration, centrifugation and concentration steps, and the material was used for SDS PAGE and immunoblots analysis, only shown in the suppl. part. What source of VLP was utilized for TEM analysis? The values detected by NTA are massively overestimated I guess, since in our lab we do not rely on NTA quantification: e.g. an HIV1-VLP sample contained >1012 particles (NTA) but less than 1010 were detected by ELISA. So, I don’t have confidence that plasmid based production of VLP yields higher titers in insect than in mammalian cells.

Spike protein and GFP expression analysis in Fig 1 shows convincing data of VLP receptor binding in the sandwich ELISA. Only construct M6-Furin has background signal. In contrast, in the cytometry assay M6-Furin does not differ so much from M1 given the extremely low dynamic range. Is there any explanation for this?

In quest of VLP real shape and morphology, cryo-EM should be used rather than TEM which produces lots of artifacts. The size of scale bars in TEM figures (Fig 2A) is not readable, at least in the manuscript copy that is provided, and is not mentioned either in the legend. Y-axes of histograms in Fig 2B are not at the same scale and need to be adjusted. Stability of the VLP was monitored after storage at different temperatures over four weeks (line359). Were they stored in cell supernatant or as sucrose concentrated samples? In the section of antibody and sera testing, the legend of Fig 5 (line 416) must be adapted (same color in text and Fig).

As always there are merits and downsides in that work. The inhibition assay is build up on transfected reporter cells (HEK293) that show high batch to batch variability when generated by plasmid transfection. Insect cells are an additional component of the assay. They serve for the production of VLP which is also based on plasmid transfection and is influenced by varying culture conditions and passaging effects. In terms of standardization, it can be challenging to obtain material of comparable quality. Aside from that, low signal to background ratios adversely affect accuracy.

On the other hand, one advantage is, the assay does not require purification of VLP and no additional antibody staining steps and can potentially be scaled to an automated format for evaluation of the neutralization potential of respective antibodies.
